# Diagnostic and Prognostic Utilities of Pancreatic Stone Protein in Patients with Suspected Sepsis

**DOI:** 10.3390/diagnostics14182076

**Published:** 2024-09-19

**Authors:** Gun-Hyuk Lee, Hanah Kim, Hee-Won Moon, Yeo-Min Yun, Mikyoung Park, Seungho Lee, Mina Hur

**Affiliations:** 1Department of Laboratory Medicine, Konkuk University School of Medicine, Seoul 05030, Republic of Korea; leegunhyuk93@gmail.com (G.-H.L.); md.hkim@gmail.com (H.K.); hannasis@hanmail.net (H.-W.M.); yun7640@gmail.com (Y.-M.Y.); 2Department of Laboratory Medicine, Ajou University School of Medicine, Suwon 16499, Republic of Korea; mikyoung.pak@gmail.com; 3Department of Preventive Medicine, College of Medicine, Dong-A University, Busan 49201, Republic of Korea; lgydr1@gmail.com

**Keywords:** sepsis, SOFA score, procalcitonin, pancreatic stone protein, diagnosis, prognosis

## Abstract

**Background/Objectives:** Pancreatic stone protein (PSP) is an emerging biomarker of sepsis that is secreted from pancreas sensing remote organ damages. We explored the diagnostic and prognostic utilities of PSP in patients with suspected sepsis. **Methods:** In a total of 285 patients (suspected sepsis, *n* = 148; sepsis, *n* = 137), we compared PSP with procalcitonin (PCT) and sequential organ failure assessment (SOFA) score. Sepsis diagnoses were explored using receiver operating characteristic curve analyses with area under the curves (AUCs). Clinical outcomes (in-hospital mortality, 30-day mortality, and kidney replacement therapy [KRT]) were explored using the Kaplan–Meier method and a multivariate analysis with hazard ratio (HR). **Results:** PCT and PSP were comparable for sepsis diagnosis (AUC = 0.71–0.72, *p* < 0.001). The sepsis proportion was significantly higher when both biomarkers increased than when either one or both biomarkers did not increase (89.0% vs. 21.3–47.7%, *p* < 0.001). Each biomarker quartile (Q1–Q4) differed significantly according to their SOFA score (all *p* < 0.001). Compared with Q1, the Q2–Q4 groups showed worse clinical outcomes (*p* = 0.002–0.041). Both biomarkers added to the SOFA score showed higher HRs than the SOFA score alone (3.3–9.6 vs. 2.8–4.2, *p* < 0.001–0.011), with nearly 2.5-fold higher HR (9.6 vs. 4.2) for predicting KRT. **Conclusions:** Although PCT and PSP did not independently predict clinical outcomes in the multivariate analysis, PSP demonstrated diagnostic and prognostic utilities in patients with suspected sepsis, especially for predicting kidney dysfunction. PSP, alone or in combination with PCT, would be a valuable tool that can be added to clinical assessments.

## 1. Introduction

Sepsis is the leading cause of death, representing 19.7% (95% uncertainty interval, 18.2–21.4) of all global deaths [1]. Due to its clinical importance, there have been many efforts to make an early diagnosis and predict prognosis in sepsis. At present, the global consensus for sepsis diagnosis is the sepsis-3 criteria, which is based on the sequential organ failure assessment (SOFA) score [2]. SOFA score is a complex scoring system calculated by the summation of six organ category scores (respiration, coagulation, liver, cardiovascular, central nervous system, and renal system). Accordingly, clinicians cannot calculate SOFA score immediately, resulting in delayed diagnosis of sepsis. Furthermore, each of the six organ categories used in the SOFA score cannot perfectly divide the risk of mortality rate in sepsis, and some clinical or laboratory parameters used in the SOFA score (mean arterial pressure, creatinine, etc.) are not perfect for evaluating each organ category [2,3,4]. With these acknowledged limitations, there has been an ongoing need to find more objective and reliable biomarkers to substitute or modify the SOFA score [5].

Procalcitonin (PCT), as a conventional biomarker and the only US Food and Drug Administration-approved biomarker for sepsis diagnosis, has shown diagnostic, prognostic, and therapeutic (regarding antibiotic stewardship) utilities in sepsis [6]. PCT has also shown increased diagnostic and prognostic performances when used together with other biomarkers or added on top of SOFA score in sepsis [7,8,9,10,11].

Pancreatic stone protein (PSP) is an emerging biomarker for sepsis, and its diagnostic and prognostic utilities have been explored in sepsis [12]. However, to our knowledge, there has been no study that simultaneously explored the utility of PSP in combination with PCT and SOFA score. PSP was firstly recognized in pancreatic juice as lithostathine and regenerating protein 1 (Reg I) in the 1980s [13]. PSP, a low-molecular-weight protein (about 14 kDa), is structurally similar to C-type lectin-like proteins that are related to inflammatory responses [14]. In sepsis, pancreas sense remote organ damages and secrete PSP without damaging pancreatic tissue, which makes PSP as an acute-phase protein [15,16]. Furthermore, PSP levels have increased in patients with kidney dysfunction, such as in patients with diabetes kidney disease or reduced estimated glomerular filtration rate (eGFR) [14,17,18]. These results suggest that PSP might be used for predicting clinical outcomes, especially for predicting kidney dysfunction in sepsis.

In this study, we aimed to explore the diagnostic and prognostic utilities of PSP in patients with suspected sepsis, in comparison with PCT and clinical assessment. Focused on a comparative analysis, diagnostic utility was evaluated with predicting sepsis status (suspected sepsis vs. sepsis), and prognostic utility was evaluated with predicting presence or absence of clinical outcomes (intensive care unit [ICU] admission, in-hospital mortality, 30-day mortality, vasopressor use, and kidney replacement therapy [KRT]). We wanted to know whether PSP would play a role in the context of a multi-marker approach in sepsis. We hypothesized that using a combination of biomarkers (PCT and PSP) or biomarkers added to the SOFA score would show improved diagnostic and prognostic performances than using the SOFA score alone. We also hypothesized that PSP could stratify the severity of kidney dysfunction in patients with suspected sepsis.

## 2. Materials and Methods

### 2.1. Study Population

From June 2020 to June 2021, a total of 483 patients were consecutively admitted to the Konkuk University Medical Center (KUMC) with clinical suspicion of developing sepsis. For all of these patients, routine laboratory testing, including C-reactive protein (CRP) and PCT, was conducted, and they were potentially enrolled according to the National Institute for Health and Care Excellence (NICE) guideline with a CRP level of 10 mg/dL or higher [19,20]. After excluding 198 patients who were younger than 20 years old or had inadequate or insufficient residual samples (residual plasma volume < 1 mL, hemolysis, or clot), we finally enrolled 285 patients in this study. At enrollment, they were divided into two groups, sepsis and suspected sepsis, according to the sepsis-3 criteria [2]; 137 patients (48.1%) were confirmed as having sepsis, and the others who did not meet the sepsis-3 criteria were considered as having suspected sepsis. Their medical records were reviewed retrospectively to retrieve demographic, clinical, and laboratory findings (Table 1). In the study population, no patient was diagnosed as having Coronavirus disease 2019 (COVID-19), which might have affected the prognosis of sepsis [21]. This was a cross-sectional, in vitro evaluation study, which consisted of forward sample collection and a retrospective analysis of the clinical and laboratory findings. The Institutional Review Board of KUMC approved the study protocol (approval No. 2023-05-073). This study used anonymized clinical data and required neither additional sampling nor intervention; accordingly, obtaining written informed consent from the study population was waived.

### 2.2. Measurement of PSP Level

Following routine testing, the residual plasma samples were aliquoted and stored at −70 °C until use. The frozen samples were thawed at room temperature and gently mixed immediately before measuring PSP level. PSP level was measured using the in vitro diagnostics (IVD) CAPSULE PSP assay (Abionic SA, Epalinges, Switzerland) on an abioSCOPE system (Abionic SA). It is a nanofluidic point-of-care immunoassay. Briefly, each sample (50 μL) was mixed with an abioMIX reagent (50 μL, composed of fluorescently labelled anti-human PSP antibody) and immediately loaded onto the PSP capsule. The PSP–antibody complex was captured by antibodies immobilized on the capsule’s read-out area. The concentration of the captured PSP was proportional to the fluorescence generated by the fluorophore conjugated to the detection antibody; therefore, the measured fluorescence signal was proportional to the PSP level in the sample. The manufacturer-suggested measurement range for PSP is 20–600 ng/mL [22]. For the internal quality control, the IVD CAPSULE control PSP (control material, REF P02.00040) was tested each day of assay.

### 2.3. Statistical Analysis

The data are presented as number (percentage) or median (interquartile range, IQR). The normality of data distribution was assessed using the Shapiro–Wilk test, and no outlier was detected using Grubb’s test. A Mann–Whitney U test or Fisher’s exact test was used to compare the SOFA score, biomarker levels, and clinical outcomes between the two groups of suspected sepsis and sepsis. For sepsis diagnosis, receiver operating characteristic (ROC) curves of PCT, PSP, and their combination were analyzed, and the DeLong test was used to compare each area under the curve (AUC) with 95% confidence interval (CI) [23]. The correlation between the PCT and PSP levels was accessed using Pearson’s correlation coefficient (r) [24]. Based on optimal cut-offs derived from the ROC curves, the distribution of PCT and PSP levels was compared using a Chi-squared test. Both the PCT and PSP levels were divided into quartiles (Q1–Q4), and the Chi-squared test was also used to evaluate the distribution of biomarker quartiles according to the SOFA score groups (five groups: 0–1, 2–3, 4–5, 6–7, and >8) and the presence or absence of clinical outcomes.

For predicting clinical outcomes (in-hospital mortality, 30-day mortality, and KRT), the Kaplan–Meier method was used for SOFA score, PCT, PSP, and their different combinations; SOFA score > 1, PCT > Q1, and PSP > Q1 were used as respective cut-offs for dichotomizing continuous variables. The log-rank test was used for survival curve comparison, and HR (95% CI) was calculated. The sample size for the Kaplan–Meier method was estimated based on a previous study [25]. The inputs were as follows: analysis time t = 1 or 8 months, accrual time α = 1 or 8 months, follow-up time b = 1 or 8 months, null survival probability S0(t) = 0.244, and alternative survival probability S1(t) = 0.123 or 0.147 [26]. Using arcsine square-root transformation and a two-sided type I error rate (α) of 0.05 and a power (1 − β) of 0.8, the estimated sample size was 130 or 169. Accordingly, the sample size was considered sufficient to apply the Kaplan–Meier method. A COX proportional hazard (PH) regression analysis was further performed to determine the effect of each biomarker on the in-hospital mortality, 30-day mortality, and KRT. Model 1 included PCT, Model 2 included PSP, Model 3 included both PCT and PSP variables, and Model 4 included whether PCT and PSP were above the first quartile. All models were adjusted for age, sex, comorbidities, and SOFA score. The PH assumption was checked in each model to assess whether the hazard associated with covariates remained constant over time; the interaction term between log (time-to-event) and covariates was used to evaluate whether the covariate on hazard ratio is constant or not over time [27].

The Kruskal–Wallis test with a post hoc analysis (Dunn) was used to evaluate the distribution of biomarkers according to the GFR categories. The GFR categories were assigned according to the Kidney Disease Improving Global Outcomes (KDIGO) 2012 guidelines: G1, ≥90 mL/min/1.73 m^2^; G2, 60–89 mL/min/1.73 m^2^; G3a, 45–59 mL/min/1.73 m^2^; G3b, 30–44 mL/min/1.73 m^2^; G4, 15–29 mL/min/1.73 m^2^; and G5, <15 mL/min/1.73 m^2^ [28]. For eGFR calculation, the 2021 chronic kidney disease epidemiology collaboration creatinine (CKD-EPICr) equation was used [29].

For the statistical analyses, MedCalc Software (version 22.009, MedCalc Software, Ostend, Belgium), R version 4.3.1, and R studio 2023.06.0+421 (The R Foundation for Statistical Computing, Vienna, Austria) were used. A *p* value < 0.05 was considered statistically significant.

## 3. Results

In a total of 285 patients, 137 patients (48.1%) were diagnosed as having sepsis (suspected sepsis, *n* = 148 vs. sepsis, *n* = 137) (Table 1). In 137 sepsis patients, 62 patients (45.3%) had community-acquired sepsis, which started within ≤72 h of hospital admittance without recent exposure to healthcare risks [30]. Median values of SOFA score and both biomarkers (PCT and PSP) differed significantly between the two groups of suspected sepsis and sepsis (all *p* < 0.001). Regarding clinical outcomes, the proportions of ICU admission, in-hospital mortality, 30-day mortality, vasopressor use, and KRT differed significantly between the two groups (all *p* < 0.001).

Both PCT and PSP could discriminate sepsis from suspected sepsis comparably with respective cut-offs of 0.41 and 306 ng/mL (AUC, 0.71 and 0.72; both *p* < 0.001); their combination also showed a comparable performance (AUC = 0.71, *p* < 0.001) (Figure 1a). There was no significant correlation between PCT and PSP levels (*r* = 0.10, *p* = 0.100) in the study population (Figure 1b). The proportion of sepsis was significantly higher when both biomarker levels increased above optimal cut-offs than when either one or both biomarker levels did not increase (89.0% vs. 21.3% to 47.7%, *p* < 0.001) (Figure 1c).

The distribution of PCT or PSP quartiles differed significantly according to SOFA score group (all *p* < 0.001). Regarding PCT, although the proportion of Q1 decreased as the SOFA scores increased (from 29.8% to 4.8%), such a trend was not observed for a SOFA score of 6–7 (17.4%) (Figure 2a). Regarding PSP, the proportion of Q1 showed a stepwise decrease as the SOFA score increased (from 35.1% to 4.8%) (Figure 2b).

The distribution of PCT or PSP quartiles (Q1 vs. Q2–Q4) was explored according to the clinical outcomes (Table 2). Although there was no significant difference regarding ICU admission and vasopressor use, the distribution of PCT or PSP quartiles differed significantly according to the three clinical outcomes (in-hospital mortality, 30-day mortality, and KRT). The proportion of PCT Q1 was significantly lower than the proportion of PCT Q2–Q4 (1.5–4.5% vs. 9.2–17.9%, *p* = 0.007–0.036), and the proportion of PSP Q1 was significantly lower than the proportion of PSP Q2–Q4 (1.4–5.7% vs. 9.3–17.7%, *p* = 0.014–0.029). The combination of PCT and PSP also showed such a finding; the proportion of biomarker quartiles was significantly higher when both biomarker levels increased to Q2–Q4 than when both biomarker levels were Q1 (10.9–20.6% vs. 0.0–3.8%, *p* = 0.002–0.041).

In the Kaplan–Meier method, SOFA score predicted in-hospital mortality, 30-day mortality, and KRT, with HRs (95% CI) of 2.9 (1.5–5.7), 2.8 (1.1–6.7), and 4.2 (1.8–9.8), respectively. The HRs of PCT, PSP, or their combination were comparable with those of the SOFA score for predicting these clinical outcomes. Of note, the PCT and PSP combination added to the SOFA score showed higher HRs than the SOFA score alone (3.3 vs. 2.9 for in-hospital mortality; 4.4 vs. 2.8 for 30-day mortality; 9.6 vs. 4.2 for KRT) (Figure 3).

The overall distribution of PCT and PSP levels differed significantly according to GFR category (both *p* < 0.001). However, the PCT levels showed overlapping IQR between G2 and G4, while the PSP levels showed no overlapping IQR between G2 and G4 (Figure 4).

In the Cox PH regression analysis, marginal significance for in-hospital mortality was observed in Model 4, which examined whether PCT and PSP were above Q1 (*p* = 0.0566). However, the hazard associated with covariates did not remain constant over time; the interaction term showed significant *p* values except SOFA score for predicting in-hospital mortality, 30-day mortality, and KRT (Table 3). These findings suggested that the PH assumption was not met in the models for predicting in-hospital mortality, 30-day mortality, and KRT [31].

## 4. Discussion

To our knowledge, this is the first study that explored the diagnostic and prognostic utilities of PSP simultaneously with PCT and SOFA score in patients with suspected sepsis. Our hypothesis was that using a combination of biomarkers (PCT and PSP) or biomarkers added to SOFA score would show improved diagnostic and prognostic performances than using SOFA score alone. We also hypothesized that PSP could stratify the severity of kidney dysfunction in patients with suspected sepsis. Our study demonstrated that PSP could discriminate sepsis from suspected sepsis and the performance of PSP was comparable with that of PCT (Figure 1a). Sepsis is a complex inflammatory pathophysiologic process; accordingly, a single biomarker used alone may not be enough for assessing organ dysfunctions, stratifying disease severity, or predicting clinical outcomes [32]. Previous studies have explored the utility of biomarker combinations or add-on values of each biomarker on top of clinical assessment, such as SOFA score, in sepsis [7,8,9,10,11,33]. In our study, there was no significant correlation between PSP level and PCT level, suggesting their independent pathophysiology in sepsis (Figure 1b). Furthermore, the proportion of sepsis was approximately 90%, when both PCT and PSP levels increased. On the contrary, the proportion of sepsis was approximately 20%, when both biomarker levels did not increase (Figure 1c). These findings suggest that PSP has its own diagnostic utility for discriminating sepsis from suspected sepsis, which is differentiated from that of PCT, and the combination of PSP and PCT would be a more comprehensive and valuable tool than the use of each biomarker alone.

In terms of risk stratification, the proportions of PCT Q1 and PSP Q1 were approximately 6.0-fold higher and 7.0-fold higher, respectively, when the SOFA score was 0–1 than when the SOFA score was more than 8 (29.8% vs. 4.8% in PCT; 35.1% vs. 4.8% in PSP) (Figure 2). These remarkable differences suggest that both PCT or PSP levels could reflect and stratify the severity of sepsis. In addition, the proportions of clinical outcomes (in-hospital mortality, 30-day mortality, and KRT) were significantly different, showing at least 5.0-fold higher proportions when both biomarker quartiles were Q2–Q4 than when both biomarker quartiles were Q1 (Table 2). The combination of PCT and PSP showed comparable performances with SOFA score for predicting these clinical outcomes. Of note, the PCT and PSP combination added on top of SOFA score showed better prognostic performances than SOFA score alone: in particular, the HR showed a nearly 2.5-fold difference for predicting KRT (HR = 9.6 in SOFA + PCT + PSP vs. HR = 4.2 in SOFA) (Figure 3). These results suggest the utility of PSP for predicting in-hospital mortality, 30-day mortality, and KRT in patients with suspected sepsis. However, both PCT and PSP did not independently predict clinical outcomes in the multivariate analysis (Table 3).

In our study, another noticeable finding was that PSP showed an add-on value on top of SOFA score, especially for predicting kidney dysfunction. In sepsis, kidney dysfunction may result in KRT and high mortality (about 40–50%); accordingly, its early detection is clinically important [34]. Based on our data, although the PCT level differed significantly according to GFR categories, the IQR of PCT levels overlapped across all GFR categories (Figure 4a). On the contrary, the PSP level was nicely distributed according to GFR categories, showing no overlapped median (IQR) values between G2 and G4 (G2, 163 [111–280] ng/mL; G4, 541 [427–600] ng/mL) (Figure 4b). These results suggest that PSP could be useful in stratifying the risk of kidney dysfunction in patients with suspected sepsis. Decreased renal blood flow is known to be the prevailing pathophysiology of kidney dysfunction in sepsis [34], and PSP can cross glomerular basement membranes and undergo reabsorption in the proximal renal tubules [35]. PSP seems to reflect a reduced glomerular filtration capacity rather than reabsorption from damaged renal tubules. In pregnancy, GFR may increase with remaining reabsorption in the proximal renal tubules, resulting in slightly lower PSP levels than in the general population [36,37]. It has been also reported that the PSP level was higher in pregnancy with reduced GFR than in pregnancy with normal GFR [18]. Considering the mechanism of PSP secretion in kidney and our data together, PSP might be beneficial for predicting kidney dysfunction in sepsis.

In previous studies, PSP has been explored mostly using isoform-specific enzyme-linked immunosorbent assay (ELISA), which is a research-use only (RUO) assay [12,17,37,38,39], and there have been only a few studies that measured PSP level using nanofluidic point-of-care IVD immunoassays [40,41,42]. Compared with ELISA, which uses serum samples, the point-of-care IVD immunoassay uses plasma samples and has advantages in terms of timely and convenient accessibility. Previous studies have also shown some differences in the clinical cut-offs and distributions of PSP level, according to the type of sample and assay. The serum PSP level measured using RUO ELISA was approximately 4.6-fold lower than the plasma PSP level measured using an IVD immunoassay, and the cut-offs for sepsis diagnosis also showed a similar difference between the serum and plasma samples [43]. In a previous study, serum PSP showed variable cut-off ranges of 29–97 ng/mL, and plasma PSP showed a cut-off of 291 ng/mL, for predicting sepsis diagnosis [12]. In non-survivors of all-cause mortality, which also included other critically ill patients other than sepsis, median levels of serum PSP varied within the range of 117–604 ng/mL, and the median level of plasma PSP was 141 ng/mL [12]. In the present study, the cut-off of PSP for sepsis diagnosis was 306 ng/mL, and the median level of PSP in non-survivors (*n* = 42) was 289 ng/mL. Because biomarker levels may be affected by various causes (sampling time, trauma, infection without sepsis, comorbidities, etc.), variable results can be obtained in different studies and needed to be assessed with careful clinical consideration [32]. The correlation between serum and plasma PSP levels should be explored and validated in further studies.

This study has several limitations. First, the PCT and PSP levels were evaluated only at enrollment of the study population. Given that sepsis may show diverse clinical courses, evaluating serial changes in both PCT and PSP levels would be more useful for assessing their clinical utilities [40,42]. Second, only the presence or absence of KRT was evaluated retrospectively for investigating the prognostic utility of PSP regarding kidney dysfunction. Further prospective studies would be needed for evaluating the prognostic utility of PSP in relation to the development of sepsis-associated acute kidney injury. Third, the study population had a skewed distribution towards the elderly (*p* < 0.001), which is an independent factor for predicting in-hospital mortality in critically ill patients such as septic patients [44]. Accordingly, our study population cannot represent the entire age group, and our results cannot be extrapolated to the other age groups. Further comprehensive evaluation including pediatric patients would be necessary to support our findings. Fourth, although the manufacturer-suggested measurement range for plasma PSP level was 20–600 ng/mL, 56 patients (19.6%) showed PSP levels higher than 600 ng/mL, and those PSP levels were assumed and analyzed as 600 ng/mL. Accordingly, it is possible that the utility of PSP might have been underestimated in our study; widening the upper measurement range of plasma PSP levels would be needed for evaluating its level precisely. Fifth, this study focused on a comparative analysis and used the Kaplan–Meier method to highlight these unadjusted trends with straightforward comparison. Further studies on developing a predictive model to validate our findings would confirm the clinical value of this multi-marker approach [45].

In conclusion, our study demonstrated that an emerging biomarker of PSP has diagnostic and prognostic utilities in patients with suspected sepsis, especially for predicting kidney dysfunction. Although PCT and PSP did not independently predict clinical outcomes in multivariate analysis, PSP, alone or in combination with PCT, would be a valuable clinical tool that can be used in conjunction with clinical assessments. Further studies are awaited to explore the pathophysiology and clinical utility of PSP in various clinical settings and to validate and extend our findings.

## Figures and Tables

**Figure 1 diagnostics-14-02076-f001:**
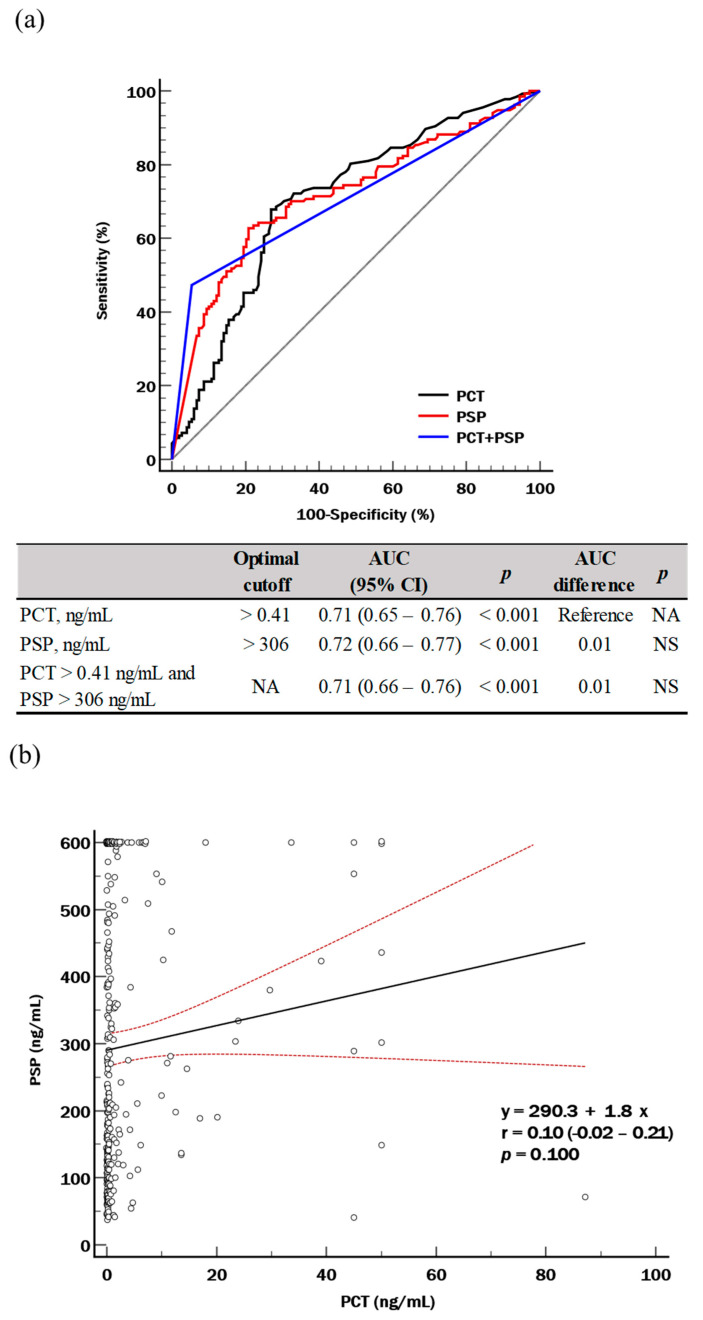
PCT and PSP for the sepsis diagnosis (suspected sepsis, *n* = 148 vs. sepsis, *n* = 137). (**a**) ROC curve analysis. (**b**) Pearson correlation coefficient analysis. (**c**) Scatter plot using above-optimal cut-off values; the biomarker levels are represented as logarithmic transformation. Abbreviations: ROC, receiver operating characteristics; AUC, area under the curve; CI, confidence interval; PCT, procalcitonin; PSP, pancreatic stone protein; NA, not available; NS, not significant.

**Figure 2 diagnostics-14-02076-f002:**
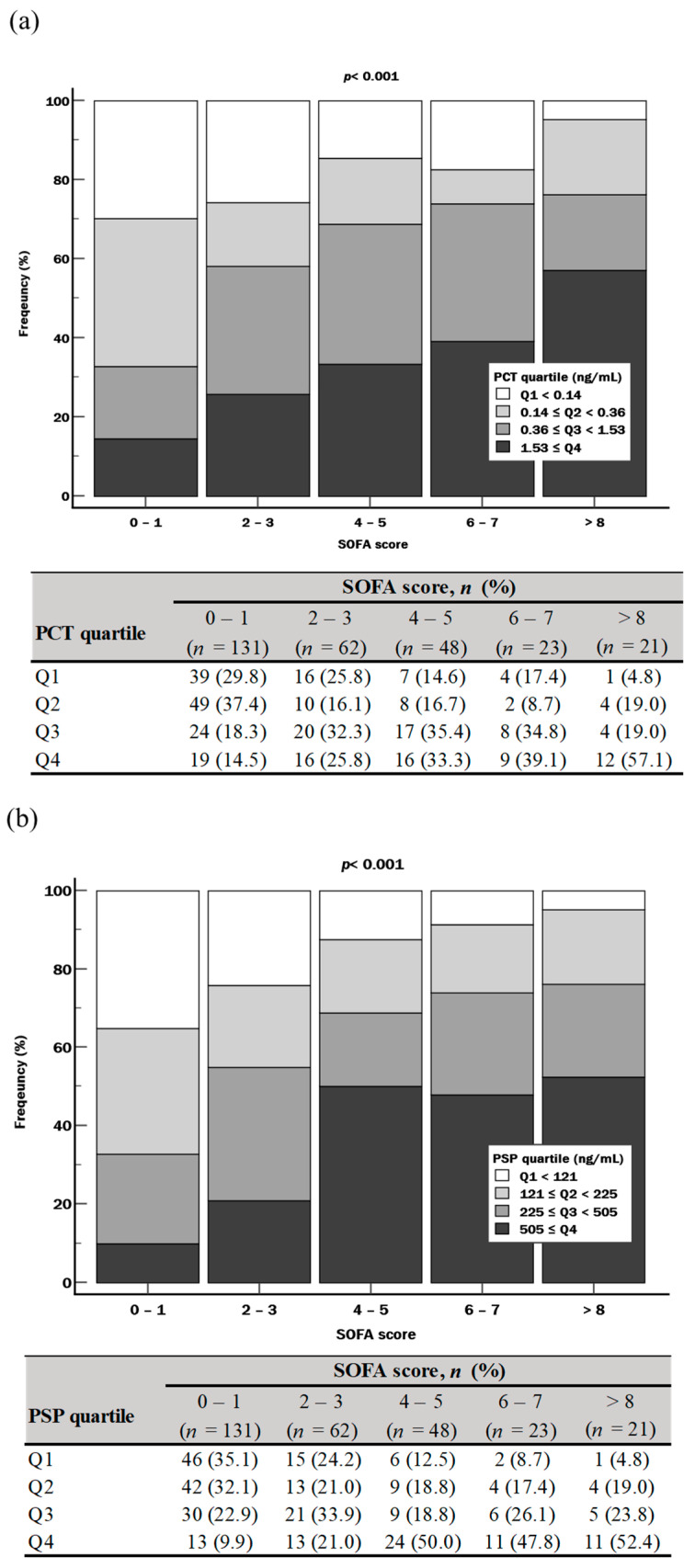
Distribution of biomarker quartiles according to SOFA score (*n* = 285). (**a**) PCT and SOFA score. (**b**) PSP and SOFA score. Abbreviations: PCT, procalcitonin; PSP, pancreatic stone protein; SOFA, sequential organ failure assessment.

**Figure 3 diagnostics-14-02076-f003:**
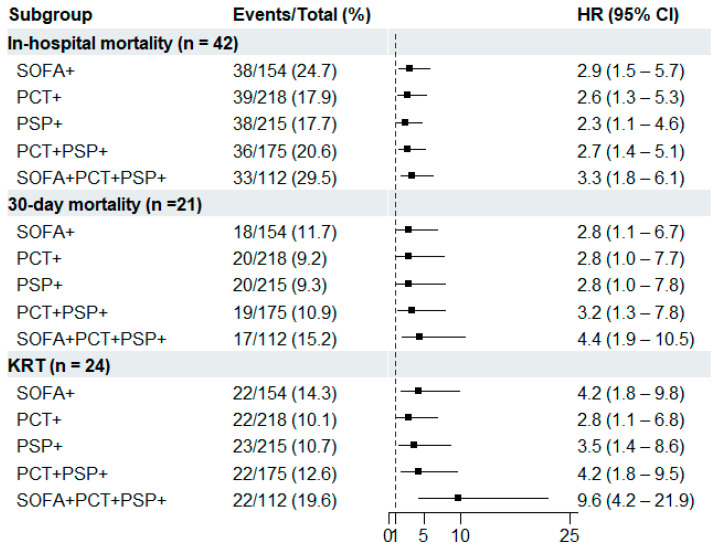
Kaplan–Meier method for SOFA score and biomarkers for predicting clinical outcomes (*n* = 285). Abbreviations: HR, hazard ratio; CI, confidence interval; PCT+, procalcitonin > Q1; PSP+, pancreatic stone protein > Q1; SOFA+, sequential organ failure assessment score > 1.

**Figure 4 diagnostics-14-02076-f004:**
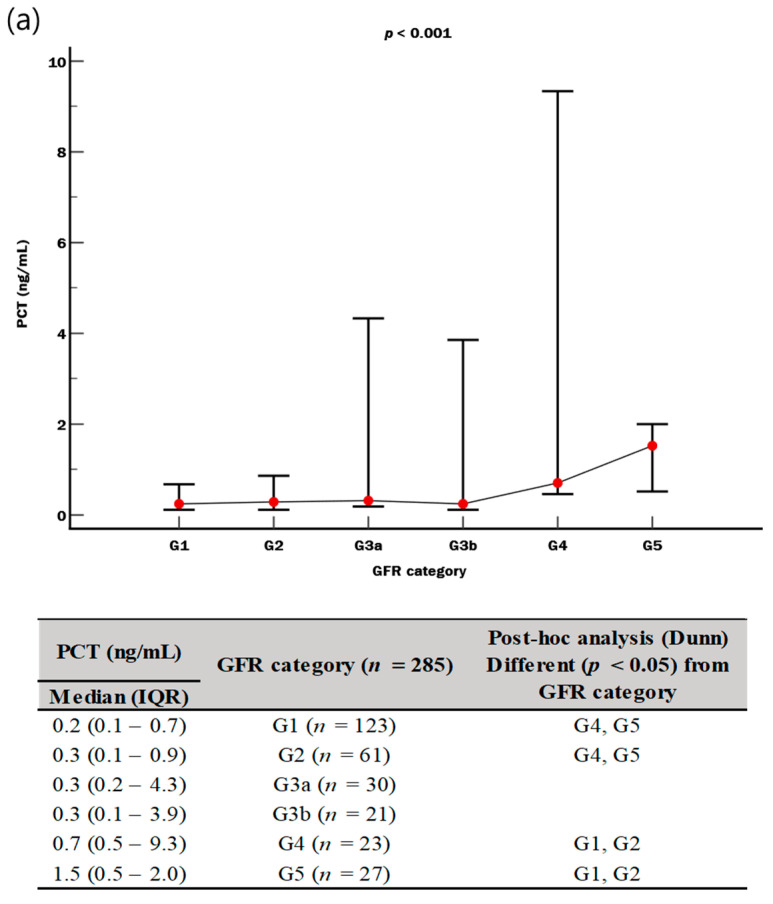
Distribution of biomarkers according to GFR category (*n* = 285). (**a**) PCT and GFR category. (**b**) PSP and GFR category. Abbreviations: IQR, interquartile range; GFR, glomerular filtration rate; PCT, procalcitonin; PSP, pancreatic stone protein.

**Table 1 diagnostics-14-02076-t001:** Basic characteristics of the study population.

	Total (*n* = 285)	Suspected Sepsis (*n* = 148)	Sepsis * (*n* = 137)	*p*
**Demographics**				
Age (yr)	68 (59–79)	64 (53–75)	74 (64–83)	<0.001
Female, *n* (%)	114 (40.0)	64 (43.2)	50 (36.5)	NS
**Comorbidities, *n* (%)**				
Malignancy	125 (43.9)	65 (43.9)	60 (43.8)	NS
Hypertension	121 (42.5)	52 (35.1)	69 (50.4)	0.012
Diabetes mellitus	89 (31.2)	41 (27.7)	48 (35.0)	NS
Chronic kidney disease	43 (15.1)	8 (5.4)	35 (25.5)	0.012
Cerebrovascular accident	39 (13.7)	9 (6.1)	30 (21.9)	<0.001
Chronic heart failure	25 (8.8)	7 (4.7)	18 (13.1)	0.020
Dementia	19 (6.7)	4 (2.7)	15 (10.9)	0.001
AIDS	2 (0.7)	2 (1.3)	0 (0.0)	NS
**Patient enrollment ****				
Admission to sampling (day)	3 (1–12)	3 (1–9)	5 (1–21)	0.011
Postoperative state, *n* (%)	6 (2.1)	0 (0.0)	6 (4.3)	NA
General ward, *n* (%)	239 (83.9)	132 (89.7)	107 (78.1)	<0.001
Emergency room, *n* (%)	27 (16.1)	14 (9.5)	13 (9.5)	NS
Intensive care unit, *n* (%)	19 (6.7)	2 (1.4)	17 (12.4)	<0.001
**SOFA score at enrollment**				
Total	2 (0–4)	1 (0–1)	4 (3–6)	<0.001
Central nervous system	0 (0–1)	0 (0–0)	1 (0–2)	<0.001
Renal	0 (0–1)	0 (0–0)	1 (0–2)	<0.001
Respiratory	0 (0–1)	0 (0–0)	1 (0–1)	<0.001
Coagulation	0 (0–0)	0 (0–0)	0 (0–1)	<0.001
Circulatory	0 (0–0)	0 (0–0)	0 (0–0)	<0.001
Liver	0 (0–0)	0 (0–0)	0 (0–1)	<0.001
**Clinical outcomes**				
ICU admission, *n* (%)	87 (29.8)	29 (19.6)	52 (40.9)	<0.001
ICU stay (day)	4 (2–19)	2 (1–6)	5 (2–34)	NS
Hospital stay (day)	20 (10–46)	13 (7–27)	29 (13–54)	<0.001
In-hospital mortality, *n* (%)	42 (14.7)	7 (4.7)	35 (25.5)	<0.001
30-day mortality, *n* (%)	21 (7.4)	5 (3.4)	16 (11.7)	<0.001
Vasopressor use, *n* (%)	34 (11.9)	6 (4.1)	28 (20.4)	<0.001
KRT, *n* (%)	24 (8.4)	1 (0.7)	23 (18.1)	<0.001
**Laboratory values at enrollment**				
WBC (×10^9^/L)	10.6 (7.8–14.4)	10.2 (7.4–14.1)	11.0 (8.6–14.9)	NS
Hb (g/dL)	10.1 (9.0–11.7)	10.5 (9.3–12.7)	10.0 (8.7–10.9)	<0.001
PLT (×10^9^/L)	224 (157–289)	249 (193–319)	187 (128–259)	<0.001
Lactate (mmol/L) ***	1.8 (1.3–2.5)	1.7 (1.2–2.3)	1.8 (1.4–2.8)	NS
Total bilirubin (umol/L)	0.7 (0.5–1.1)	0.6 (0.4–0.9)	0.8 (0.5–1.5)	<0.001
Cr (μmol/L)	0.9 (0.6–1.3)	0.8 (0.6–1.0)	1.3 (0.7–2.4)	<0.001
eGFR (mL/min/1.73 m^2^)	83 (46–101)	93 (70–104)	55 (20–94)	<0.001
CRP (mg/L)	15.9 (12.5–21.3)	15.7 (12.2–22.1)	15.8 (12.5–21.0)	NS
PCT (ng/mL)	0.36 (0.14–1.53)	0.20 (0.11–0.50)	0.60 (0.26–2.50)	<0.001
PSP (ng/mL)	225 (121–506)	164 (101–280)	408 (162–730)	<0.001

The data are presented as number (percentage) or median (interquartile range). * In the 137 sepsis patients, 128 patients showed culture-positivity (*S. aureus*, *n* = 27; *E. coli*, *n* = 19; *E. faecium*, *n* = 11; *K. pneumonia*, *n* = 11; *A. baumannii*, *n* = 9; *S. epidermidis*, *n* = 7; *P. aeruginosa*, *n* = 6; *C. utilis*, *n* = 5; *C. albicans*, *n* = 4; *E. fecalis*, *n* = 3; *K. aerogenes*, *n* = 3; *S. marcescens*, *n* = 3; *S. maltophilia*, *n* = 3; *E. cloacae*, *n* = 2; others, *n* = 15) and 9 patients had radiographic findings suggestive of infection. ** The reasons for initial hospital admission of the 137 sepsis patients were pneumonia, *n* = 43; solid tumor, *n* = 33; urinary tract infection, *n* = 20; biliary tract infection, *n* = 12; soft tissue infection, *n* = 7; cerebrovascular accident, *n* = 4; congestive heart failure, *n* = 3; infective endocarditis, *n* = 3; liver abscess, *n* = 2; and others, *n* = 10. *** Lactate level was measured in 210 patients (suspected sepsis, *n* = 88; sepsis, *n* = 122). Abbreviations: NS, not significant; NA, not available; AIDS, acquired immune deficiency syndrome; ICU, intensive care unit; KRT, kidney replacement therapy; SOFA, sequential organ failure assessment; eGFR, estimated glomerular filtration rate; PCT, procalcitonin; PSP, pancreatic stone protein.

**Table 2 diagnostics-14-02076-t002:** Distribution of biomarker quartiles according to clinical outcomes.

	PCT Quartile, *n* (%)			PSP Quartile, *n* (%)			PCT and PSP Combination, *n* (%)				
	PCT Q1 (*n* = 67)	PCT Q2–Q4 (*n* = 218)	*p*	PSP Q1 (*n* = 70)	PSP Q2–Q4 (*n* = 215)	*p*	PCT Q1 PSP Q1 (*n* = 27)	PCT Q2–Q4 PSP Q1 (*n* = 43)	PCT Q1 PSP Q2–Q4 (*n* = 40)	PCT Q2–Q4 PSP Q2–Q4 (*n* = 175)	*p*
ICU admission (*n* = 87)											
No	43 (64.2)	155 (71.1)	-	53 (75.7)	145 (67.4)	-	20 (74.1)	33 (76.7)	23 (57.5)	122 (69.7)	-
Yes	24 (35.8)	63 (28.9)	NS	17 (24.3)	70 (32.6)	NS	7 (25.9)	10 (23.3)	17 (42.5)	53 (30.3)	NS
In-hospital mortality (*n* = 42)											
No	64 (95.5)	179 (82.1)	-	66 (94.3)	177 (82.3)	-	25 (96.2)	40 (93.0)	38 (95.0)	139 (79.4)	-
Yes	3 (4.5)	39 (17.9)	0.007	4 (5.7)	38 (17.7)	0.014	1 (3.8)	3 (7.0)	2 (5.0)	36 (20.6)	0.002
30-day mortality (*n* = 21)											
No	66 (98.5)	198 (90.8)	-	69 (98.6)	195 (90.7)	-	27 (100.0)	42 (97.7)	39 (97.5)	156 (89.1)	-
Yes	1 (1.5)	20 (9.2)	0.036	1 (1.4)	20 (9.3)	0.029	0 (0.0)	1 (2.3)	1 (2.5)	19 (10.9)	0.041
Vasopressor use (*n* = 34)											
No	56 (83.6)	195 (89.4)	-	64 (91.4)	187 (87.0)	-	23 (85.2)	41 (95.3)	33 (82.5)	154 (88.0)	-
Yes	11 (16.4)	23 (10.6)	NS	6 (8.6)	28 (13.0)	NS	4 (14.8)	2 (4.7)	7 (17.5)	21 (12.0)	NS
KRT (*n* = 24)											
No	66 (98.5)	195 (89.4)	-	69 (98.6)	192 (89.3)	-	26 (96.3)	43 (100.0)	40 (100.0)	152 (86.9)	-
Yes	1 (1.5)	23 (10.6)	0.020	1 (1.4)	23 (10.7)	0.016	1 (3.7)	0 (0.0)	0 (0.0)	23 (13.1)	0.003

Abbreviations: NS, not significant; ICU, intensive care unit; KRT, kidney replacement therapy; PCT, procalcitonin; PSP, pancreatic stone protein.

**Table 3 diagnostics-14-02076-t003:** Cox proportional hazard regression analysis for predicting adverse clinical outcomes (*n* = 285).

* Variable	Model 1			Model 2			Model 3			Model 4		
	OR	95% CI	*p*	OR	95% CI	*p*	OR	95% CI	*p*	OR	95% CI	*p*
No in-hospital mortality (*n* = 243) vs. In-hospital mortality (*n* = 42)												
Age > 65	NS	NS	NS	-	-	-	NS	NS	NS	NS	NS	NS
Male	NS	NS	NS	NS	NS	NS	NS	NS	NS	NS	NS	NS
Comorbidity number > 1	NS	NS	NS	-	-	-	NS	NS	NS	NS	NS	NS
SOFA score > 1	3.676	1.279–10.562	0.016	3.787	1.191–9.941	0.022	3.466	1.197–10.031	0.022	3.441	1.191–9.941	0.022
PCT > Q1	NS	NS	NS	-	-	-	NS	NS	NS	-	-	-
PSP > Q1	-	-	-	NS	NS	NS	NS	NS	NS	-	-	-
PCT > Q1 and PSP > Q1	-	-	-	-	-	-	-	-	-	NS	NS	NS
log(t)*(age > 65)	-	-	-	-	-	-	-	-	-	0.209	0.072–0.609	0.004
log(t)*(male)	-	-	-	-	-	-	-	-	-	0.078	0.015–0.406	0.002
log(t)*(comorbidity number > 1)	-	-	-	-	-	-	-	-	-	0.021	0.009–0.683	0.021
log(t)*(SOFA score > 1)	-	-	-	-	-	-	-	-	-	NS	NS	NS
log(t)*(PCT > Q1 and PSP > Q1)	-	-	-	-	-	-	-	-	-	<0.001	<0.001–0.001	<0.001
No 30-day mortality (*n* = 264) vs. 30-day mortality (*n* = 21)												
Age > 65	NS	NS	NS	-	-	-	NS	NS	NS	NS	NS	NS
Male	NS	NS	NS	NS	NS	NS	NS	NS	NS	NS	NS	NS
Comorbidity number > 1	NS	NS	NS	-	-	-	NS	NS	NS	NS	NS	NS
SOFA score > 1	6.209	1.699–22.695	0.006	6.342	1.782–22.568	0.004	6.155	1.700–22.279	0.006	6.209	1.699–22.695	0.006
PCT > Q1	NS	NS	NS	-	-	-	NS	NS	NS	-	-	-
PSP > Q1	-	-	-	NS	NS	NS	NS	NS	NS	-	-	-
PCT > Q1 and PSP > Q1	-	-	-	-	-	-	-	-	-	NS	NS	NS
log(t)*(age > 65)	-	-	-	-	-	-	-	-	-	0.057	0.013–0.242	0.004
log(t)*(male)	-	-	-	-	-	-	-	-	-	0.033	0.001–0.753	0.002
log(t)*(comorbidity number > 1)	-	-	-	-	-	-	-	-	-	<0.001	0.002–0.130	0.021
log(t)*(SOFA score > 1)	-	-	-	-	-	-	-	-	-	NS	NS	NS
log(t)*(PCT > Q1 and PSP > Q1)	-	-	-	-	-	-	-	-	-	0.002	<0.001–0.001	<0.001
No KRT (*n* = 261) vs. KRT (*n* = 24)												
Age > 65	NS	NS	NS	-	-	-	NS	NS	NS	NS	NS	NS
Male	0.224	0.066–0.757	0.016	0.223	0.066–0.752	0.016	0.225	0.067–0.756	0.016	0.225	0.067–0.756	0.016
Comorbidity number > 1	NS	NS	NS	-	-	-	NS	NS	NS	NS	NS	NS
SOFA score > 1	11.894	1.580–89.540	0.016	12.424	1.659–93.074	0.014	11.249	1.494–84.673	0.019	11.894	1.580–89.540	0.016
PCT > Q1	NS	NS	NS	-	-	-	NS	NS	NS	-	-	-
PSP > Q1	-	-	-	NS	NS	NS	NS	NS	NS	-	-	-
PCT > Q1 and PSP > Q1	-	-	-	-	-	-	-	-	-	NS	NS	NS
log(t)*(age > 65)	-	-	-	-	-	-	-	-	-	0.020	0.023–0.165	0.005
log(t)*(male)	-	-	-	-	-	-	-	-	-	0.008	0.006–0.107	0.008
log(t)*(comorbidity number > 1)	-	-	-	-	-	-	-	-	-	0.043	0.002–0.656	0.021
log(t)*(SOFA score > 1)	-	-	-	-	-	-	-	-	-	NS	NS	NS
log(t)*(PCT > Q1 and PSP > Q1)	-	-	-	-	-	-	-	-	-	<0.001	<0.001–0.001	<0.001

* Used covariates were as follows: Model 1, age, sex, comorbidity, SOFA score, and PCT; Model 2, age, sex, comorbidity, SOFA score, and PSP; Model 3, age, sex, comorbidity, SOFA score, PCT, and PSP; and Model 4, age, sex, comorbidity, SOFA score, and the PCT and PSP combination. Abbreviations: NS, not significant; SOFA, sequential organ failure assessment; PCT, procalcitonin; PSP, pancreatic stone protein; Q1, quartile 1; log(t), log (time to event).

## Data Availability

The data presented in this study are available from the corresponding author on reasonable request.

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
