# Peer review of "Diagnostic and Prognostic Utilities of Pancreatic Stone Protein in Patients with Suspected Sepsis"

_diagnostics, 2024, doi:10.3390/diagnostics14182076_

Round 1

Reviewer 1 Report

Comments and Suggestions for Authors

PSP has been reported previously for diagnosis of sepsis. This report is trying to combine PSP, PCT and SOFA scores in the diagnosis of sepsis. However, this study lack proper multiple variate analysis and have not eliminated confounding factors to demonstrate those variates that variates are independent factors. In addition, the authors have not designed testing cohort and validation cohort. No validation data confirm the values of this combination in the diagnosis of sepsis.

The tables and figures shall be more standardized. 

Comments on the Quality of English Language

Moderate correction

Author Response

  1. PSP has been reported previously for diagnosis of sepsis. This report is trying to combine PSP, PCT and SOFA scores in the diagnosis of sepsis. However, this study lack proper multiple variate analysis and have not eliminated confounding factors to demonstrate those variates that variates are independent factors.

Thank you for your comment. This study focused on the comparative analysis of adverse clinical outcomes between the groups with different biomarker levels (presence vs. absence). The Kaplan-Meier method was an effectively way to show these differences, allowing for a straightforward comparison without the complexity of multivariate adjustments. Given the goal of our study and according to your valuable opinion, we added the following sentence in the Discussion section as one of the study limitations. (page 13)

Fifth, this study focused on comparative analysis and used Kaplan-Meier method to high-light these unadjusted trends with straightforward comparison. Further studies including multivariate analysis to explore potential confounding factors and developing a predictive model to validate our findings would confirm the clinical value of this multi-marker approach.

  1. In addition, the authors have not designed testing cohort and validation cohort. No validation data confirm the values of this combination in the diagnosis of sepsis.

Thank you for your comments. This study aimed to evaluate the diagnostic and prognostic utility of PSP within a well-defined patient population based on clinical diagnosis (suspected sepsis) We focused on the comparative analysis to assess the utility of PSP between groups (sepsis vs. suspected sepsis, absence of adverse clinical outcomes vs. presence of adverse clinical outcomes) rather than developing a predictive model that would necessitate separate validation and testing cohorts. Given the goal of our study and according to your valuable opinion, we added and modified the following sentences in Introduction and Discussion sections.

Focused on the comparative analysis, diagnostic utility was evaluated with predicting sepsis status (suspected sepsis vs. sepsis), and prognostic utility was evaluated with pre-dicting presence or absence of clinical outcomes (intensive care unit [ICU] admission, in-hospital mortality, 30-day mortality, vasopressor use, and kidney replacement therapy [KRT]). We wanted to know whether PSP would play a role in the context of multi-marker approach in sepsis. (page 2)

Fifth, this study focused on comparative analysis and used Kaplan-Meier method to high-light these unadjusted trends with straightforward comparison. Further studies including multivariate analysis to explore potential confounding factors and developing a predictive model to validate our findings would confirm the clinical value of this multi-marker approach. (page 13)

  1. The tables and figures shall be more standardized.

Thank you for your comment. According to your comment, we modified Tables and Figures.

Reviewer 2 Report

Comments and Suggestions for Authors

I read with interest paper entitled „Diagnostic and prognostic utilities of pancreatic stone protein in patients with suspected sepsis“. The authors explored if combination of biomarkers procalcitonin and PSP or biomarkers added on SOFA score would improve diagnostic and prognostic performances than using SOFA score alone in detecting sepsis. The paper addresses an interesting research question and, data are logically presented and overall is nicely written.

However, there are several issues that authors should address to improve the manuscript:

1.       Study design should be described in more detail.
Was this a prospective study?
Add inclusion and exclusion criteria.
Define “suspected sepsis” vs “sepsis”.

2.       Results – data on sepsis source and etiology should be provided. These could influence the patients’ clinical course, survival and levels of inflammatory markers.
Were only patients with community acquired sepsis included? What about postoperative and already hospitalized patients or immunocompromised? Were they included? They could have different PSP levels?  

3.       Discussion should be rewritten – First three paragraphs describe main study findings. It is reasonable that in the first paragraph the authors highlight their main findings or introduce the reader to the discussion with a short paragraph in the context of the proposed issue. Afterward, compare with previous results in the literature. Point out the similarities and differences between your work and the manuscripts analyzed, discuss the probable mechanisms, and make a hypothesis of the obtained results.

Comments on the Quality of English Language

Minor editing required

Author Response

I read with interest paper entitled „Diagnostic and prognostic utilities of pancreatic stone protein in patients with suspected sepsis“. The authors explored if combination of biomarkers procalcitonin and PSP or biomarkers added on SOFA score would improve diagnostic and prognostic performances than using SOFA score alone in detecting sepsis. The paper addresses an interesting research question and, data are logically presented and overall is nicely written.

However, there are several issues that authors should address to improve the manuscript:

  1. Study design should be described in more detail. Was this a prospective study? Add inclusion and exclusion criteria. Define “suspected sepsis” vs “sepsis”.

Thank you for your comment. According to your comment, we added and modified the following sentences in the Methods section (study population). (page 2) We also added two more references (19 and 20).

From June 2020 to June 2021, a total of 483 patients was consecutively admitted to the Konkuk University Medical Center (KUMC) with clinical suspicion of developing sepsis. For all of these patients, routine laboratory testing, including C-reactive protein (CRP) and PCT, was conducted, and they were potentially enrolled according to the National Institute for Health and Care Excellence (NICE) guideline with CRP level of 10 mg/dL or higher [19, 20]. After excluding 198 patients who were younger than 20 years old or had inade-quate or insufficient residual samples (residual plasma volume < 1 mL, hemolysis, or clot), we finally enrolled 285 patients in this study. At enrollment, they were divided into two groups of sepsis and suspected sepsis, according to the sepsis-3 criteria [2]; 137 patients (48.1%) were confirmed as having sepsis, and the others who do not meet the sepsis-3 criteria were considered as having suspected sepsis. Their medical records were reviewed retrospectively to retrieve demographic, clinical, and laboratory findings (Table 1). In the study population, no patient was diagnosed as having Coronavirus disease 2019 (COVID-19), which might affect the prognosis of sepsis [21]. This was a cross-sectional, in vitro evaluation study, which consisted of forward sample collection and retrospective analysis of clinical and laboratory findings.

  1. National Institute for Health and Care Excellence (NICE). QuikRead go for C-reactive protein testing in primary care 2019. Available online: https://www.nice.org.uk/advice/mib78 (accessed on 2 Jan 2024).
  2. National Institute for Health and Care Excellence (NICE). Sepsis: recognition, diagnosis and early management. Available online: https://www.nice.org.uk/guidance/ng51 (accessed on 2 Jan 2024).

  1. Results – data on sepsis source and etiology should be provided. These could influence the patients’ clinical course, survival and levels of inflammatory markers. Were only patients with community acquired sepsis included? What about postoperative and already hospitalized patients or immunocompromised? Were they included? They could have different PSP levels?

Thank you for your comment. According to your comment, we added the following sentence in the Results section and modified Table 1. There were two AIDS patients and six postoperative patients. Although we included it in Table 1, the PSP levels could not be compared due to the small sample size. We also added a new reference (reference 29).

In 137 sepsis patients, 62 patients (45.3%) had community-acquired sepsis, which started within ≤ 72 h of hospital admittance without recent exposure to healthcare risks [29]. (page 3)

  1. Szabo, B.G.; Kiss, R.; Lenart, K.S.; Marosi, B.; Vad, E.; Lakatos, B.; et al. Clinical and microbiological characteristics and out-comes of community-acquired sepsis among adults: a single center, 1-year retrospective observational cohort study from Hungary. BMC Infect Dis 2019, 19, 584.

*In 137 sepsis patients, 128 patients showed culture-positivity (bacteremia, n = 46; respiratory, n = 42; urinary, n = 20; wound, n = 10; abdominal, n = 10), and nine patients had radiographic findings suggestive of infection. (Table 1, footnote)

  1. Discussion should be rewritten – First three paragraphs describe main study findings. It is reasonable that in the first paragraph the authors highlight their main findings or introduce the reader to the discussion with a short paragraph in the context of the proposed issue. Afterward, compare with previous results in the literature. Point out the similarities and differences between your work and the manuscripts analyzed, discuss the probable mechanisms, and make a hypothesis of the obtained results.

Thank you for your comment. According to your valuable comment, we added/modified the following sentences in the Discussion section.

Our hypothesis was that using combination of biomarkers (PCT and PSP) or biomarkers added on SOFA score would show improved diagnostic and prognostic performances than using SOFA score alone. We also hypothesized that PSP could stratify the severity of kidney dysfunction in patients with suspected sepsis. (page 11)

Considering the mechanism of PSP secretion in kidney and our data together, PSP might be beneficial for predicting kidney dysfunction in sepsis. (page 12)

Round 2

Reviewer 1 Report

Comments and Suggestions for Authors

Multiple variant analysis for this type of study is essential. The factors, such as age, sex, underlying diseases, PCT, PSP and SOFA scores, shall be included in the analysis and find out the independent ones. Analysis is just one day's work. Without this analysis, the whole paper sounds odd. Best do it and present it properly.

The validation cohort is under the guide of TRIPODS statement (https://bmjopen.bmj.com/content/10/9/e041537). However, it can be done in future work at the moment.

Author Response

  1. Multiple variant analysis for this type of study is essential. The factors, such as age, sex, underlying diseases, PCT, PSP and SOFA scores, shall be included in the analysis and find out the independent ones. Analysis is just one day's work. Without this analysis, the whole paper sounds odd. Best do it and present it properly.

Thank you for your valuable comment and we apologize for our incomplete response during the first revision. According to your comment, we performed multiple variant analysis using covariates (age, sex, comorbidity, SOFA score, PCT, PSP) using 4 different models. We also evaluated interaction term between log(time-to-event) and covariates to check the COX proportional hazards (PH) assumption. We added the following sentences in the Methods and Results sections (page 3 and page 11). We also added two more references (reference 27 and 31). Although we did not include the data in the manuscript, we added the following Tables for your reference. Please, accept our reply.

COX proportional hazards (PH) regression analysis was further performed to determine the effect of each biomarker on the in-hospital mortality, 30-day mortality, and KRT. Mod-el 1 included PCT, Model 2 included PSP, Model 3 included both PCT and PSP variables, and Model 4 included whether PCT and PSP were above the first quartile. All models were adjusted for age, sex, comorbidities, and SOFA score. The PH assumption was checked in each model to assess whether the hazard associated with covariates remained constant over time; interaction term between log (time-to-event) and covariates were used to evaluate whether covariate on hazard ratio is constant or not over time [27]. (Page 3)

In the Cox PH regression analysis, marginal significance for in-hospital mortality was observed in Model 4, which examined whether PCT and PSP were above Q1 (p = 0.0566). However, the hazard associated with covariates did not remain constant over time; the interaction term showed significant p values except SOFA score for predicting in-hospital mortality, 30-day mortality, and KRT (data not shown). These findings suggested that the PH assumption was not met in the models for predicting in-hospital mortality, 30-day mortality, and KRT [31]. (Page 11)

  1. Deo, S. V.; Deo, V.; Sundaram, V. Survival analysis—part 2: Cox proportional hazards model. Indian J Thorac Cardiovasc Surg 2021, 37, 229–233.
  1. The validation cohort is under the guide of TRIPODS statement (https://bmjopen.bmj.com/content/10/9/e041537). However, it can be done in future work at the moment.

Thank you for your comment. We added the TRIPODS statement as a reference.

Further studies on developing a predictive model to validate our findings would confirm the clinical value of this multi-marker approach [45].

  1. Zamanipoor Najafabadi, A.H.; Ramspek, C.L.; Dekker, F.W.; Heus, P.; Hooft, L.; Moons, K.G.M.; et al. TRIPOD statement: a preliminary pre-post analysis of reporting and methods of prediction models. BMJ Open 2020, 10, e041537.

  1. Kuitunen, I.; Ponkilainen, V.T.; Uimonen, M.M.; Eskelinen, A.; Reito, A. Testing the proportional hazards assumption in cox regression and dealing with possible non-proportionality in total joint arthroplasty research: methodological perspectives and review. BMC Musculoskelet Disord 2021, 22, 489.

Reviewer 2 Report

Comments and Suggestions for Authors

The authors have sufficiently responded to all comments and significantly improved the manuscript. However, additional data should be provided:

1.       45% of patients were diagnosed with community acquired sepsis? This means that 55% had healthcare-associated or nosocomial infection? The reasons for initial hospital admission should be provided.

2.       Page 5: “*In 137 sepsis patients, 128 patients showed culture-positivity (bacteremia, n = 46; respiratory, n = 42; urinary, n = 20; wound, n = 10; abdominal, n = 10), and nine patients had radiographic findings suggestive of infection”
This is confusing and should be rephrased. Culture positivity means that pathogen is detected and identified (S.aureus, S. pneumoniae, E. coli….), and if available this data should be provided. The data in parentheses probably mean source of infection.  

Comments on the Quality of English Language

Minor editing required. 

Author Response

The authors have sufficiently responded to all comments and significantly improved the manuscript. However, additional data should be provided:

  1. 45% of patients were diagnosed with community acquired sepsis? This means that 55% had healthcare-associated or nosocomial infection? The reasons for initial hospital admission should be provided.

Thank you for your comment. According to your comment, we added the following sentence as a footnote of Table 1 (Page 5).

**The reasons for initial hospital admission of 137 sepsis patients were: pneumonia, n = 43; solid tumor, n = 33; urinary tract infection, n = 20; biliary tract infection, n = 12; soft tissue infection, n = 7; cerebrovascular accident, n = 4; congestive heart failure, n = 3; infective endocarditis, n = 3; liver abscess, n = 2; others, n = 10.

2.Page 5: “*In 137 sepsis patients, 128 patients showed culture-positivity (bacteremia, n = 46; respiratory, n = 42; urinary, n = 20; wound, n = 10; abdominal, n = 10), and nine patients had radiographic findings suggestive of infection”

This is confusing and should be rephrased. Culture positivity means that pathogen is detected and identified (S.aureus, S. pneumoniae, E. coli….), and if available this data should be provided. The data in parentheses probably mean source of infection.

Thank you for your comment. According to your comment, we rephrased the following sentence in the footnote of Table 1.

*In 137 sepsis patients, 128 patients showed culture-positivity (S.aureus, n = 27; E.coli, n = 19; E.faecium, n = 11; K.pneumonia, n = 11; A.baumannii, n = 9; S.epidermidis, n = 7; P.aeruginosa, n = 6; C.utilis, n = 5; C.albicans, n = 4; E. fecalis, n = 3; K.aerogenes, n = 3; S.marcescens n = 3; S.maltophilia, n = 3; E.cloacae, n = 2; others, n = 15), and nine patients had radiographic findings suggestive of infection.

Round 3

Reviewer 1 Report

Comments and Suggestions for Authors

The regression analysis showed that the biomarker is not independently predicting mortality, but good for diagnosis and prediction of kidney injury. Those points shall be reflected in Abstract and final conclusion. A table of the results of regression analysis shall be presented.

Author Response

  1. The regression analysis showed that the biomarker is not independently predicting mortality, but good for diagnosis and prediction of kidney injury. Those points shall be reflected in Abstract and final conclusion.

Thank you for your comments. According to your comment, we modified the following sentences in Abstract and Discussion (Conclusion).

Clinical outcomes (in-hospital mortality, 30-day mortality, and kidney replacement therapy [KRT]) were explored using Kaplan-Meier method and multivariate analysis with hazard ratio (HR). (Abstract)

Although PCT and PSP did not independently predict clinical outcomes in multivariate analysis, PSP demonstrated diagnostic and prognostic utilities in patients with suspected sepsis, especially for predicting kidney dysfunction. (Abstract)

However, both PCT and PSP did not independently predict clinical outcomes in multi-variate analysis (Table 3). (page 14)

Although PCT and PSP did not independently predict clinical outcomes in multivariate analysis, PSP, alone or in combination with PCT, would be a valuable clinical tool that can be used in conjunction with clinical assessments. (page 15)

  1. A table of the results of regression analysis shall be presented

Thank you for your comments. According to your comment, we added a new Table (Table 3).

In the Cox PH regression analysis, marginal significance for in-hospital mortality was observed in Model 4, which examined whether PCT and PSP were above Q1 (p = 0.0566). However, the hazard associated with covariates did not remain constant over time; the interaction term showed significant p values except SOFA score for predicting in-hospital mortality, 30-day mortality, and KRT (Table 3). (Page 11)